# Research Progress of Arc Additive Manufacture Technology

**DOI:** 10.3390/ma14061415

**Published:** 2021-03-15

**Authors:** Dan Liu, Boyoung Lee, Aleksandr Babkin, Yunlong Chang

**Affiliations:** 1School of Material Science and Engineering, Shenyang University of Technology, Shenyang 110870, China; liudan-0828@163.com; 2Department of Materials, Liaoning Mechatronics College, Dandong 118000, China; 3School of Aerospace and Mechanical Engineering, Korea Aerospace University, Seoul 101-601, Korea; bylee@kau.ac.kr; 4Institute of Mechanical Engineering, Lipetsk State Technical University, Lipetsk 398024, Russia; bas-43@yandex.ru

**Keywords:** Arc additive manufacturing technology, rapid prototyping, residual stress, pores, mechanical properties

## Abstract

Additive manufacturing technology is a special processing technology that has developed rapidly in the past 30 years. The materials used are divided into powder and wire. Additive manufacturing technology using wire as the material has the advantages of high deposition rate, uniform composition, and high density. It has received increasingly more attention, especially for the high efficiency and rapid prototyping of large-size and complex-shaped components. Wire arc additive manufacturing has its unique advantages. The concept, connotation, and development history of arc additive manufacturing technology in foreign countries are reviewed, and the current research status of arc-based metal additive manufacturing technology is reviewed from the principles, development history, process, and practical application of arc additive manufacturing technology. It focuses on the forming system, forming material, residual stress and pores, and other defect controls of the technology, as well as the current methods of mechanical properties and process quality improvement, and the development prospects of arc additive manufacturing technology are prospected. The results show that the related research work of wire arc additive manufacturing technology is still mainly focused on the experimental research stage and has yet not gone deep into the exploration of the forming mechanism. The research work in this field should be more in-depth and systematic from the physical process of forming the molten pool system from the perspectives of stability, the organization evolution law, and performance optimization. We strive to carry out wire arc additive forming technology and theoretical research to promote the application of this technology in modern manufacturing.

## 1. Introduction

Additive manufacturing (AM) technology is based on the discrete-stacking principle, which uses the method of stacking materials layer by layer to manufacture the required parts. Compared with traditional equivalent material manufacturing (casting, forging, etc.) technology and subtractive manufacturing (turning, milling, etc.) technology, additive manufacturing technology focuses on computer control of the manufacturing process and refines the manufacturing process to any part. Any point in the location is a revolutionary breakthrough in manufacturing. Additive manufacturing technology can effectively reduce processes and shorten the production cycle of products. For products with complex shapes and high raw material value, the fast and efficient production and processing characteristics of additive manufacturing technology are particularly obvious. In the fields of biomedicine and aerospace, additive manufacturing technology has shown very broad prospects.

Wire arc additive manufacturing (WAAM) technology uses electric arcs as a heat source to melt the welding wire, adopts the principle of layer-by-layer cladding under program control, and is an advanced digital manufacturing technology that gradually forms a line-surface-body based on a three-dimensional digital model. The advantages of low technical cost, high production efficiency, and simple equipment have become an important means to realize economic and rapid prototyping of metal parts. The formed parts are composed of full welds, with flexible forming and uniform chemical composition. However, because the WAAM process is carried out in a high-temperature liquid metal transition-forming method, there are problems such as the accumulation of heat input and difficulty in controlling the shape and boundary. These problems seriously restrict the surface quality, dimensional accuracy, and mechanical properties of WAAM parts.

This article expounds the domestic and foreign research and application status of WAAM technology from several major perspectives such as the control of defects such as the forming system, forming material, residual stress and pores, as well as mechanical properties and process quality improvement. Finally, research on WAAM technology is carried out. Looking forward, an overview of the current domestic and foreign WAAM technology research is shown in Table 1.

## 2. Wire Arc Additive Manufacturing (WAAM) Systems

### 2.1. Classification of WAAM Process

WAAM adopts the heat source of gas tungsten arc welding (GTAW), gas metal arc welding (GMAW), or plasma arc welding (PAW). It uses the principle of layer-by-layer deposition, and through the addition of wire, the welding wire is melted layer by layer. The 3D products produced by the upper surfacing welding process is shown in Figure 1. The workpiece formed by WAAM technology is composed of full weld metal with uniform composition and high density. Compared with other additive manufacturing technologies, WAAM technology does not require a vacuum environment and expensive equipment, and can be manufactured using conventional welding equipment. Due to the advantages of high material utilization, low equipment cost, high deposition rate, and easy real-time repair, WAAM technology has attracted increasingly more attention from domestic and foreign researchers. However, the arc additive manufacturing has poor forming accuracy and generally requires secondary surface machining.

#### 2.1.1. Based on GMAW

The gas metal arc welding (GMAW) additive manufacturing technology uses metal welding wire as the melting electrode under shielding gas, and generates an arc between the welding wire and the weldment to melt the welding wire into the molten pool for rapid forming. GMAW additive manufacturing is shown in Figure 2.

Xiong et al. proposed a laser vision-based system to observe the surface morphology of one side of a multi-layer single-channel thin-walled low-carbon-steel workpiece formed by GMAW additive manufacturing technology, and established a corresponding evaluation method to quantify the surface roughness, which is used to study the influence of process parameters on the side surface roughness of thin-walled parts [10].

In order to prove the feasibility of using fusion welding for rapid prototyping, Ribeiro and others combined CAD software and GMAW technology to “print” an all-metal vase, proposed the welding parameters needed for rapid prototyping using a welding robot, and designed a set robot rapid prototyping system [11,12].

Zhang et al. addressed the problem that the height difference between the arc starting end and the arc extinguishing end gradually increased with the increase in the number of stacked layers during GMAW additive forming, and proposed the arc starting end, the arc extinguishing end, and the droplet transition. The method control measures and the comparison before and after technical improvement are shown in Figure 3. It can be seen that before the technical improvement, when a certain number of layers were stacked, the forming process could not continue, but after the technical improvement, the forming effect significantly improved [13].

#### 2.1.2. Based on GTAW

Gas tungsten arc welding (GTAW) additive manufacturing technology is similar to GMAW technology. The difference is that the electrodes of GTAW technology are made of tungsten with a high melting point, so it is also called tungsten inert gas protection (TIG) welding. Compared with GMAW, GTAW technology has a lower energy input, is not easy to splash, and has a more stable structure and better forming effect. Therefore, this technology has been widely used in precision manufacturing fields such as aerospace. The principle of typical GTAW surfacing welding is shown in Figure 4.

Ouyang et al. manufactured 5356 aluminum alloy parts through variable-polarity GTAW additive manufacturing technology. The study found that the key to improving the forming quality of parts is to control the arc length and substrate preheating temperature: Preheat the substrate to 118 °C, adjust the arc current during the deposition process, and monitor the arc length, which effectively improves the wettability and deformation tendency of the deposited layer [14]. Subsequently, Ouyang et al. optimized the process parameters of the GTAW technology and formed a 4034 aluminum alloy hollow cylindrical part with a stack of 120 layers and good surface quality [15]. Gault et al. used Ti-6Al-4V titanium alloy as raw material and used the GTAW additive manufacturing process to study an empirical model to predict the shrinkage, geometric shape, and surface finish of the formed part. The average relative error of the preliminary empirical model developed was less than 10%, and the model needed to be improved by adding more data to improve accuracy [16].

Ma et al. used a combination of the GTAW process and in-situ alloying method for additive manufacturing of titanium aluminide alloys, and studied the effect of interlayer temperature on forming parts. The test results showed that with the change in interlayer temperature, there was no obvious change in microstructure and tissue composition [17].

#### 2.1.3. Based on PAW

Plasma arc welding (PAW) is a fusion welding method that uses a plasma arc high-energy-density beam as a welding heat source. Compared with TIG welding, PAW welding has the advantages of concentrated arc energy, good arc stiffness, and high arc stability, but the equipment is more complicated and consumes a lot of gas. According to the mode of operation and the speed of plasma gas flow, PAW welding can be divided into: Micro-beam plasma arc welding, plasma arc welding, and pierce-through-plasma arc welding. Micro-plasma arc welding has high energy density, low heat input, and high weld forming accuracy, so it is especially suitable for precision repair of titanium alloy products.

Zhen established an additive manufacturing system based on micro-plasma arc surfacing and studied the additive manufacturing process of Ti-6Al-4V titanium alloy. After surfacing ten layers, it was found that the internal grains of the micro-plasma arc surfacing layer were the coarse columnar crystal morphology, its growth direction was perpendicular to the welding direction, the inside of the crystal grain was a mesh basket structure, and the obvious layered morphology could be observed between each layer [18].

Jhavar, S. used the micro-beam plasma arc fuse process to replace the laser additive manufacturing process to produce small parts, and obtained a more regular and smooth weld bead shape. The wall thickness could reach 2.45 mm when processing straight walls, and the maximum deposition rate was 42 g/h. There were no cracks, pores, or inclusions between the deposited layers; the comparison found that the micro-plasma arc process had a smaller heat input and could deposit smaller columnar crystals per unit length. Therefore, he pointed out that the production structure is simple for small parts, and the micro-plasma arc fuse additive technology is suitable [19].

Pulse plasma arc welding has high energy density and high stiffness. The pulse technology further improves the controllability of the heat source. Kaibo used the orthogonal test method to study the peak current, pulse frequency, duty cycle, welding speed, and wire feeding speed. The influence law of weld bead size, and the establishment of a calculation model for the height between layers of thin-walled remanufactured parts, can further improve the dimensional accuracy of formed parts [20].

### 2.2. Robotic WAAM System

Welding robots are one of the most commonly used industrial robots, widely used in the fields of automobiles, electronics, and manufacturing. The use of welding robots can achieve precise control of motion parameters and motion trajectories, and solve the complex problem of arc additive structure trajectory control. Therefore, welding robots have been attracting increasingly more attention by researchers. The robot WAAM platform is mainly composed of welding robots, welding machines, collaborative control systems, control cabinets, and welding workbenches, as shown in Figure 5.

Many research institutions are currently conducting research on robotic WAAM technology. The main types of robotic welding platforms are shown in Table 2.

It can be seen from Table 2 that the current research institutes mainly choose robots from 4 major families (KUKA, ABB, FANUC, YASAKAWA) in the construction of robot additive manufacturing platforms. On the one hand, because additive manufacturing trajectory control is more complicated and the required accuracy is correspondingly high, more mature robots are generally used for platform construction; on the other hand, because these brands are equipped with offline programming software, it is convenient for complex trajectory compilation. It can be better for the manufacture of complex contours by additive manufacturing.

### 2.3. Development of Processing Method

According to the different heat sources used, additive manufacturing technology is mainly divided into laser additive manufacturing technology, electron beam additive manufacturing technology, arc additive manufacturing technology, and metal solid phase additive technology. Among them, the arc additive manufacturing technology is a new technology first proposed by German scientists. The technology uses metal welding wire as the raw material, adopts the method of submerged arc welding, stacks the melted materials layer by layer according to the pre-designed path, and finally solidifies molding to form large-size parts.

As early as 1925, the American Baker et al. used the arc as the heat source to produce “3D-printed” metal decorative objects through the layer-by-layer deposition of metal droplets. In the 1970s, German scholars first proposed the concept of using metal welding wire as raw material to manufacture large-size metal parts by submerged arc welding. Ujiie et al. used SAW, TIG, etc. as heat sources and used different kinds of welding wires to form a pressure vessel whose outer wall was a gradient material. In 1983, German Kussmaul et al. used submerged arc welding to pile up layer by layer to manufacture large-size cylindrical thick-walled stainless steel metal containers with a total weight of 79 ton and a deposition rate of 80 kg/h, and the formed material had a high tensile strength, yield strength, and toughness. In the 1990s, the UK launched two major studies that accelerated the development of WAAM technology. One was where Ribeiro et al. described in detail the process of “rapid prototyping technology based on metal materials”; the other was where Spencer et al. fixed the GMAW welding torch to a six-axis robot, and then proceeded with the rapid manufacturing of parts. In addition, Zhang et al. also published similar work and provided a process method for manufacturing vertical wall and rotating parts. In 1993, Prinz and Weiss et al. installed welding equipment on CNC milling machines, called Shaped Metal Deposition (SMD), and applied for related patents. From 1994 to 1999, the Welding Engineering Research Centre of Cranfield University (Cranfield University) developed the forming deposition manufacturing technology (SMD) for the British aircraft engine company Rolls Royce. Instead of traditional casting technology, the performance of titanium alloys, high-temperature alloys, aluminum alloys, and other materials formed by this technology had been studied and evaluated. In 2007, Clayfield University carried out research work on WAAM technology and applied the technology to the rapid manufacturing of aircraft fuselage structures.

The arc additive manufacturing technology adopts the traditional MIG welding method, which is characterized by high heat input. During the forming process, the output heat source repeatedly moves on the newly generated and formed parts, which increases the heat accumulation and makes the material during the stacking process, and a series of problems such as splashing and formation of multiple air holes will occur.

### 2.4. Application of Composite Energy Fields

High-frequency-pulse TIG welding should give full play to the characteristics of high-frequency arcs and take into account the requirements of no noise. We recommend that the current pulse frequency adopts a rectangular wave of 20 kHz [21]. Experiments have proved that these characteristics of high-frequency arcs are related to the magnitude of the current. The smaller the current, the more obvious these characteristics are. Therefore, it is suitable for microbeam welding and microbeam TIG plasma arc welding. Generally, high-frequency TIG welding can better reflect its characteristics only when the current is less than 160 A. Especially in the welding of precision thin parts, it can give full play to its advantages.

A device for generating a transverse rotating magnetic field is designed to control the rotation speed of the rotating magnetic field and the amplitude of the magnetic induction by adjusting the frequency of the low-frequency carrier wave and the duty ratio of the high-frequency amplitude modulation wave; the device is applied to TIG welding, combined with theoretical analysis. In addition, the test method reveals the movement mechanism of the TIG welding arc under the action of the transverse rotating magnetic field; the results show that the TIG welding arc regularly performs alternating rotating and offset movements under the action of the transverse rotating magnetic field, and the arc’s radius of rotation and rotation speed are always changing [22].

Based on the GTAW (gas tungsten arc welding) power supply, the super-audio DC pulsed square wave current with di/dt ≥ 50 A/μs output pulse current is used in the 0Cr18Ni9Ti austenitic stainless steel GTAW welding process, research and analysis of welding arc characteristics, and arc force and weld penetration characteristics. The results show that the parameters of the ultra-audio DC pulse square wave current have a significant impact on the electrical characteristics, shape, and arc force. Compared with the conventional DC GTAW, the fast-changing ultra-audio DC pulse GTAW arc exhibits a significant contraction effect and weld penetration. In a certain range, the melting width decreases, the average arc force increases greatly, and the welding efficiency increases.

In order to realize the precise control of the forming and formation of thin-walled metal in additive manufacturing, based on the movement of charged particles under the action of a DC magnetic field, the arc behavior under the action of an external DC magnetic field was studied, as well as the arc deflection degree, deflection direction, and additional relationship between magnetic fields. On the basis of arc deflection, the causes of weld morphology and weld grain changes are analyzed. The results show that the arc deflection occurs under the action of an external DC magnetic field, the degree of arc deflection increases with magnetic field strength within the range of the test parameters, and the direction of arc deflection is related to the direction of the external magnetic field. When a positive DC magnetic field is applied, the molten pool is backward in the welding direction offset, the weld reinforcement is relatively increased, and the weld grains are refined compared with no external magnetic field. When the reverse DC magnetic field is applied, the molten pool is located under the arc, the weld reinforcement is relatively reduced, and the weld grains are more refined than the external magnetic field refinement. The external DC magnetic field controls the shape of the weld, which has a “shape control” effect; the external DC magnetic field has an obvious grain refinement effect, which can achieve the purpose of “control.”

The effect of the external magnetic field is one of the effective ways to affect the arc additive forming process and the performance of formed parts. In order to study the effect of the applied longitudinal steady-state magnetic field on the surface quality and mechanical properties of low-carbon-steel arc additive forming parts, a GMAW-based longitudinal steady-state magnetic field-assisted arc additive forming device was built, using morphology analysis, metallographic observation, and performance testing [23]. The method contrasts and analyzes the difference in surface quality, microstructure, and mechanical properties of formed samples with and without the action of an external magnetic field. The results show that compared with ordinary fusion deposition, under the action of an external magnetic field, the width-to-height ratio of a single weld bead is increased, forming a wide and flat cross-sectional shape of the weld bead, thereby effectively improving the lap accuracy and improving the surface quality of the deposited layer. Electromagnetic stirring can also refine crystal grains and reduce fusion defects and the uneven distribution of crystal grains in multiple overlapping areas. In addition, the effect of an external magnetic field also changes the proportion and distribution of ferrite and pearlite. The mechanical performance test shows that the magnetic field improves the mechanical performance of the formed sample in the deposition direction and the lap direction, and the anisotropy of the mechanical performance is reduced.

## 3. Metals Used in WAAM Process

### 3.1. Titanium Alloys

Titanium alloy workpieces have low density, high specific strength, and good corrosion resistance, and have been widely used in aerospace, new energy, and other fields. However, the traditional processing methods are complex, inefficient, and costly, so the processing of titanium alloy workpieces is more suitable for use in additive manufacturing.

Wang and Baufeld et al. found that the mechanical properties of WAAM titanium alloy molded parts parallel to and perpendicular to the deposition direction are anisotropic. The reason is that the primary β phase is epitaxially grown throughout the sample, resulting in the microstructure and performance in all directions. Wang et al. [24,25] summarized the reasons for the formation of anisotropy and found that when the temperature drops from the liquidus to the β solidus, the WAAM titanium alloy generates the columnar β phase, the solid phase transition occurs when the temperature continues to decrease, and the α phase (lamella, Net basket shape, needle shape, etc.) precipitates from the β phase grain boundary. During the next pass of deposition, the thick β-phase at the boundary of the molten pool serves as the nucleation site, and epitaxially grows into the molten pool before solidification. The new β-phase continues to grow on the previous β-phase, so the primary columnar β-phase runs through the entire test. This kind of epitaxial growth results in significant anisotropy in both strength and plasticity of the molded part [26].

Fencheng and others from Nanchang Hangkong University found in the arc deposition test of Ti-6Al-4V-molded parts that the microstructure of the molded parts was dominated by coarse columnar crystals grown epitaxially along the substrate, composed of α and β phases. The morphology of the α phase varied with the change in the forming position. In the upper part of the forming part, there was a thick needle-like Widmanstatten structure; in the middle part, there was a coarsened lamellar mesh basket; and in the lower part, there was a needle-like + Lamellar mixed structure. The hardness of the molded part was larger near the surface of the workpiece; the tensile properties showed obvious anisotropy, the tensile strength along the welding horizontal direction was higher than the strength along the deposition height direction, and the plasticity was the opposite [27].

It can be found that the inhomogeneity of the composition of the different regions of the microstructure of the WAAM titanium alloy moldings and the resulting anisotropy of the mechanical properties are the main reasons why its application is currently limited. Research has found that optimizing the forming process parameters and heat treatment can effectively improve the overall uniformity of the formed parts, weaken the anisotropy, and improve the comprehensive mechanical properties of the formed parts.

Jianjun et al. found in the preparation test of Ti-6Al-4V thin-walled parts that as the heat input of each weld pass during the deposition process decreased, the average yield strength (YS), tensile strength (UTS), elongation, and growth rate increased significantly [28]. Ping et al. reached a similar conclusion on the research of the TIG additive preparation of Ti-6Al-4V molded parts. As the welding current decreased and the welding speed increased, the tensile strength of the molded parts increased. As the cooling time increased, the strength and plasticity of the material improved [29]. Ning of Harbin Institute of Technology found that the increase in welding current and the decrease in welding speed will lead to the growth of the grains of the Ti-6Al-4V molded parts in arc surfacing welding, the formation of massive α phases, and even the formation of coarse clusters. Increasing the cooling time between layers and appropriately increasing the wire feeding speed can homogenize the organization, eliminate the cluster organization, and effectively reduce the material anisotropy [30]. Åkerfeldt et al. found that a faster cooling rate during the deposition process usually makes the material structure show a finer microstructure, thereby increasing the yield strength and tensile strength of Ti-6Al-4V molded parts, but reducing the plasticity [31]. Brandl et al. found that furnace cooling aging heat treatment at 600 °C for 4 h can significantly increase the hardness of Ti-6Al-4V molded parts; furnace cooling and solution treatment at 1200 °C for 2 h can make the columnar β crystal equiaxed. The band characteristics disappear and the hardness decreases [32]. Wauthle et al. found that stress-relieving heat treatment of Ti-6Al-4V molded parts can effectively promote the transformation of the α martensite into equilibrium phase α lamella, while hot isostatic pressing treatment can transform the structure into isotropic lamellar. The α + β phase of the product can effectively improve the plasticity of the molded part [33]. Brandl et al. used high-temperature solution + quenching + annealing heat treatment of titanium alloy-molded parts, and found that a large number of columnar β crystals in the structure of Ti-6Al-4V-molded parts transformed into spheres, and the material isotropy of the molded parts greatly increased [34]. The research by Zhi et al. showed that ultrasonic impact can effectively transform the columnar crystals in the structure of WAMM titanium alloy moldings into equiaxed crystals, thereby eliminating the anisotropy of the moldings and optimizing the overall performance of the moldings [35]. Bermingham et al. added a trace amount of boron during the forming process, and found that the α grain boundary and cluster structure in the formed part were significantly reduced, but the number of α equiaxed crystals increased instead, which significantly reduced the anisotropy [36].

In summary, it can be seen that in the process of arc surfacing welding Ti-6Al-4V molded parts, the increase in welding current and the decrease in welding speed will lead to an increase in the level of heat input, which promotes the growth of structure grains, resulting an increase in material strength. The level of mechanical properties such as plasticity and plasticity is reduced. It may also form a cluster structure and cause obvious anisotropy of molded parts. By appropriately reducing the current, increasing the welding speed, and extending the cooling time between layers, the grain size can be significantly refined, the cluster structure can be effectively eliminated, and the comprehensive mechanical properties of the material can be improved. At the same time, suitable heat treatments such as solution treatment, quenching, annealing, and other post-treatments have a certain effect on refining the grain size and reducing the anisotropy of the material. In short, follow-up research in this area includes how to suppress the formation of undesirable structures inside the molded parts in the arc additive preparation of titanium alloys, effectively control the evolution of grains and microstructures during the layer-by-layer accumulation process, reduce anisotropy, and improve the comprehensive mechanical properties of parts [37].

### 3.2. Aluminum Alloys and Steel

Aluminum alloy-forming materials have a series of advantages such as good fracture toughness, high specific strength, low density, and good corrosion resistance. They have become the preferred materials commonly used in arc additive manufacturing. At present, the main reason that hinders the wide application of WAAM-forming aluminum alloy components is composition segregation, pore defects, and discontinuities between the deposited layer and the bonding layer structure during the forming process.

Jiuyang et al. studied the 4043 aluminum alloy surfacing-forming parts and found that the slender columnar crystals in the structure continuously grew through the interlayer, the dendrite structure was obvious, there was a large amount of Al-Si eutectic structures in the dendrite gap, and the composition was seriously segregated [38]. When studying the pore defects in the WAAM aluminum alloy sample structure, Baoqiang et al. found that the deposited layer was composed of equiaxed grains, the pores were mainly distributed in the interlayer remelting area, and they were distributed along the equiaxed grain boundaries, which were the distinctive features. The hydrogen pore nucleation rate and its size have an important relationship with the grain size of the deposited layer structure. Grain refinement can effectively limit the diffusion of hydrogen in the liquid metal, thereby limiting the nucleation and growth of the hydrogen pores. The size and number of pores in the deposited layer are significantly reduced. In addition, increasing the flow of pure argon shielding gas also helps reduce alloy pore defects. At present, improving the structure and performance of aluminum alloy WAAM-forming parts is mainly achieved by refining the structure and grains. The more commonly used methods are optimizing welding parameters and using appropriate heat treatment [39,40]. Wanglan’s research on TIG arc additive preparation of 5356 aluminum alloy molded parts showed that the microstructure of the molded parts was a α solid solution with a large number of β (Mg5Al8) phases dispersed and distributed. The β phase can play a role in dispersion strengthening of the alloy and significantly improve the corrosion resistance and plasticity of the material. With the increase in welding current, the number of β phases in the structure decreased and the size increased, which is not conducive to the formation of a good welded structure. With the increase in welding speed and wire feeding speed, the number of β phases in the structure increased and the size decreased, which is beneficial to the formation of a good post-weld structure [41]. Jihui’s research on TIG surfacing 2219 aluminum alloy-forming parts reached a similar conclusion: Increasing the welding speed and wire feeding speed can promote the transformation of the structure from a polygonal cell crystal to cellular dendrites, reduce the grain size, and improve the material strength and plasticity [42]. Shen et al.’s research on arc surfacing of Fe3Al-based aluminum alloys showed that in order to achieve the expected accumulation structure, the complete melting and mixing of the welding wire required a sufficiently large welding current. However, as the welding current increased, the size of the epitaxial columnar crystal increased, and the plasticity of the material reduced when the current was too large [43]. Rui’s research on the structure and properties of 5356 aluminum alloy surfacing samples showed that with the increase in heat input, the structure gradually changed from uniform equiaxed crystals to coarse columnar crystals, increasing the phase precipitation, size, and impurity phases. As a result, the mechanical properties such as the microhardness and tensile strength of the material were significantly reduced; at the same time, as the waiting time between layers increased, the grains became finer and more uniform, the microstructure changed to equiaxed crystals, and the above mechanical properties improved [44].

Hongxuan et al. studied the effect of solution aging treatment on the mechanical properties of 6063 aluminum alloy and found that with the increase in solution treatment temperature, the dendrites in the alloy decreased, the composition distribution tended to be uniform, and the degree of supersaturation of the solid solution increased, which is beneficial to the mechanical properties. During the subsequent aging treatment, more strengthening phases were precipitated to increase the strength of the alloy. The morphology obtained by solution treatment at 525 °C for 4 h and aging treatment at 210 °C for 6 h was most ideal. While the solution treatment temperature continued to increase to 540 °C, it would cause coarse grains and even over-sintering, resulting in a decrease in the overall properties of the alloy [45]. Gu et al. found that with the increase in rolling load, the microhardness and strength of 2319 aluminum alloy-formed parts gradually increased; and after T6 heat treatment, the yield strength and tensile strength of the deposited layer and the interlayer rolled alloy increased [46]. Fixter et al. found that in the preparation process of 2024 aluminum alloy arc-forming parts, the porosity could be significantly reduced by rolling and refrigerating each layer, while complete heat treatment (T4, T6) could enhance the yield strength, tensile strength, and plasticity of the forming part [47].

In summary, it can be seen that the size of each phase in the structure of the arc surfacing aluminum alloy-forming part shows a coarsening trend with the increase in heat input, and the mechanical properties such as material strength and plasticity decrease. Therefore, reducing the heat input level, such as increasing the welding speed, wire feeding speed, and the waiting time between layers, or appropriately reducing the welding current, can refine the grain of the molded part and improve the mechanical properties of the material. In addition, suitable post-treatment processes such as solution aging treatment and complete annealing can also improve the structure of the molded part to varying degrees to achieve the purpose of improving the mechanical properties.

Stainless steel materials have high strength, good ductility, and good corrosion resistance, and the deposited layer structure and mechanical properties are relatively stable. In actual production, the forming method of multi-layer and multi-layer continuous accumulation can be adopted. Many experimental studies have found that by changing the welding current, welding speed, and other surfacing process parameters and proper post-weld heat treatment, the microstructure and mechanical properties of stainless steel arc-formed parts have a certain improvement.

Fencheng et al. discovered during the surfacing welding of 316 L stainless steel samples that the liquid metal was mainly dissipated through the substrate during the cooling and solidification process, and the heat was mainly dissipated downward perpendicular to the substrate. The direction of heat flow was perpendicular to the interface, and the solidification was directional, thus causing the structure to present the form of columnar crystals growing from bottom to top [48]. Zhixi and others found that with the gradual increase in welding speed during the surfacing process of stainless steel, the amount of δ ferrite in the metal microstructure of the surfacing layer increased significantly, and its morphology also changed significantly. The comparison found that the welding speed was moderate (v = 8 m/h), and a surfacing layer structure with low heat input, low dilution rate, good forming, austenite matrix, and slender continuous framework-like delta ferrite could be obtained [49]. Sadeghian et al. analyzed the microstructure of TIG stainless steel-welded joints under different heat inputs by simultaneously changing the welding current and welding speed. The sample was brittle-fracture; as the heat input increased to 861 J/mm, the percentage of ferrite decreased significantly, and perlite and ferrite phases were produced. The strength of the sample under the two heat input conditions was higher than that of the substrate metal [50]. Zhongyi and others changed the welding heat input level by increasing the welding current, and studied the relationship between the heat input and the ferrite content in the surfacing stainless steel specimens and obtained a similar conclusion. When the welding current was increased from 160 to 220 A, the ferrite content decreased from 7.2% to 5.1% [51]. In order to study the influence of heat input on the structure and performance of stainless steel-forming parts, Wang et al. prepared 304 L stainless steel thin-walled parts with heat input levels of 271 and 377 J/mm, respectively. The analysis found that the forming parts with lower linear heat input performed better. It was characterized by a refined microstructure so that the yield strength, tensile strength, and ductility levels were higher [52]. Kumar et al. reached similar conclusions on the microstructure and mechanical properties of 304 stainless steel joints of argon tungsten arc welding. The heat input level was changed mainly by adjusting the welding current, and compared with 256, 278, and 302 J/mm. Under the input conditions, the microstructure and tensile properties of the joints found that the grains in the heat-affected zone of the joints were significantly refined at low heat input levels, and the average dendrite length and dendrite spacing in the weld zone were smallest, resulting in the highest tensile strength of the sample [53]. Yu et al. found that increasing the heat input level after surfacing a certain number of layers was conducive to the complete austenitization of the previous weld, and ultimately a more uniform recrystallized structure. The welding process of the last weld seam adopted a small current and a faster welding speed, which could increase the cooling rate of the final weld seam, obtain a large amount of martensite, and ensure the edge strength and hardness of the formed part [54].

It can be found that the change in welding process parameters has an obvious influence on the microstructure and mechanical properties of stainless steel arc-forming parts. To a certain extent, the reduction in welding current and the increase in welding speed will lead to the refinement of the structure of the formed part, the reduction in dendrite length and spacing, and the increase in ferrite content. Therefore, the mechanical properties show a tendency to decrease in hardness and increase in strength and plasticity. In addition, proper heat treatment and other post-treatment methods can also improve the structure and performance of stainless steel-molded parts.

Yadollahi et al. found that the homogenization heat treatment at 1150 °C for 2 h caused the 316 L stainless steel-molded parts to coarsen the grains, increase the large-angle grain boundaries, and reduce the delta ferrite. The structure after heat treatment was almost all austenite, and the anisotropy was eliminated, the yield strength of the molded parts decreased by 17%, the tensile strength decreased by 5%, the elongation after fracture increased by 26%, and the plasticity was significantly enhanced [21]. Wang et al. studied the effect of heat treatment on the structure and properties of deposited martensitic stainless steel, and found that high-temperature solution treatment above 1000 °C for 30 min and quenching in oil caused a large number of columnar crystal structures to be transformed into equiaxed crystals, and interdendritic phases were obtained. After quenching and tempering, the tensile strength of the material could be increased by 80 MPa compared with the deposited state, the elongation after fracture could be increased by 6%, and both exceeded the forging material [55].

Numerous studies have shown that the heat input and the resulting accumulation of repeated thermal cycles have the most obvious impact on the structure and performance of the stainless steel-deposited layer. Generally, the lower part of the molded part has undergone repeated thermal cycles to present an equiaxed morphology, and the upper part has not undergone repeated heating. It is cycle and present typical weld structure. As a result, the microstructures of the upper, middle, and lower parts of the molded part show a large difference, resulting in a gradient distribution of the overall structure and properties.

### 3.3. Functionally Gradient Materials

Based on relevant research results at home and abroad, the research team analyzed the solidification mechanism of 309 L austenitic stainless steel and 5356 aluminum alloy using TIG welding (TIG) as the heat source, and discussed the relationship between the structure formation characteristics and mechanical properties of different parts. The layer-by-layer cladding forming characteristics of arc additive lead to the characteristics of alternate layers of bonding and deposition in the microstructure of the molded parts, which is often related to the evolution of the mechanical properties of the molded parts. Based on TIG surfacing welding of 309 L austenitic stainless steel-forming parts, the research team analyzed the evolution of the microstructure and mechanical properties of 309L stainless steel during arc additive forming. The test results showed that the matrix of the surfacing sample was austenite, and there are residual dendrites of ferrite.

Bin et al. studied the shielding gas suitable for 316 L wire arc additive and the formability of 316 L welding wire under different welding parameters, and determined that the shielding gas suitable for 316 L arc additive was 98%Ar + 2%O_2_, and in good shape. The range of technical parameters and organizational characteristics were unique, and qualified neck flange parts were manufactured by arc additive. Almeida et al. conducted an arc additive manufacturing test of a single wall structure of Ti-6Al-4V material.

Oguzhan used pulse-current GTAW additive manufacturing to manufacture stainless steel parts, using AISI 308 L Si stainless steel welding wire. In the study, the influence of pulse frequency and other deposition process parameters on the morphology and microstructure characteristics of additive manufacturing components was compared.

Dan et al. studied the 5A06 aluminum alloy GTA-AM process, and chose φ 1.2 mm 5A06 aluminum alloy welding wire as the forming material, using the TIG AC mode welding machine as the power source, and the four-axis linkage numerical control mechanism as the motion actuator. It is single-layer single-pass substrate preheating temperature and arc peak current process specification criteria.

Martin used ER4043 welding wire as the raw material and used the arc additive method to develop a large cone-shaped cylinder with a height of about 380 mm. Bombardier used arc additive technology to directly manufacture large aircraft ribs on large flat plates. They were about 2.5 m long and about 1.2 m wide.

## 4. Common Defects in WAAM-Fabricated Component

### 4.1. Residual Stresses and Distortion

Although there are many advantages of WAAM, there are still some defects in WAAM manufacturing that need attention and resolution.

In the WAAM process, the heat input is large, the temperature field distribution of the workpiece is complex, and the residual stress generated during the additive process seriously restricts the quality of the formed part. At present, the residual stress of additive components is mainly reduced by changing the additive method, optimizing the additive path, and performing pre-weld preheating, interlayer cooling, and post-weld heat treatment.

Longwei et al. summarized the causes of residual stress in the WAAM process. One is due to the uneven temperature field distribution and inconsistent cooling and solidification, and the other is due to the local phase transformation of the metal to produce phase transformation residual stress [56]. Szost et al. compared the microstructure and residual stress of Ti-6Al-4V alloy parts formed by WAAM and laser additive manufacturing. The results show that the residual stress in all directions of the WAAM-forming part is relatively large, and the maximum residual stress appears at the bottom of the forming layer [1]. Colegrove et al. used in-situ rolling to reduce residual stress in the process of adding TC4 titanium alloy, which can effectively reduce the peak residual stress, especially the residual stress between the bottom layer and the substrate, and refine the grain [57,58]. Rujian and others applied laser shock to improve the microstructure and mechanical properties of arc additive 2319 aluminum alloy components. The results show that after laser shock strengthening, the average grain size reduced from 68.86 μm before impact to 34.32 μm, and the microhardness increased from 67.8 HV before impact to 100.6 HV. The residual stress changed from the tensile residual stress inside the material at a depth of 0.2 mm. It became the residual compressive stress of approximately 90 MPa at the maximum impact depth of 0.65 mm [6]. Guilan et al. introduced a magnetic field in the WAAM process of low-alloy steel to improve the residual stress distribution of the formed part. The induction heat generated by the oscillating electromagnetic field on the surface of the workpiece made the temperature distribution more uniform, the cooling rate slow down, and the overall residual stress reduce [59]. Bai et al. added induction heating as the second heat source to the additive manufacturing process, and analyzed the residual stress in the three states of no second heat source, preheating in advance, and continuous heating after welding, and found that both preheating and continuous heating after welding increased the heat input and reduced the residual stress. Preheating in advance will reduce the tension mismatch and the yield stress of the material, so the preheating effect is better [60].

### 4.2. Porosity

Stomatal defects are one of the main defects of metal additive manufacturing, seriously affecting the quality of manufactured parts. Effective control or even elimination of stomatal defects is the key technology to improve additive manufacturing parts.

Cold metal transfer (CMT) is one of the most attractive additive methods in the manufacturing of metal electrode arc additives. It combines droplet transfer with wire drawing technology. During droplet transfer, the arc is extinguished and the welding current is almost zero, which greatly reduces the welding heat input and improves the deposition rate. Cong et al. first studied and found that increasing the flow of pure argon shielding gas can effectively reduce the generation of pores in aluminum alloy additive manufacturing, and then statistically classified the pore diameters under four different CMT (cold metal transfer) processes. The results showed that most of the pore diameters were 10–50 μm. Compared with the other three modes, the traditional CMT mode produced the most pores, with some pores exceeding 100 μm in diameter. The CMT + P process significantly reduced the number of pores and the size of the pores, the CMT + ADV process further reduced the number of pores, and the CMT + PADV process almost eliminated the stomata. Lei used the double pulse arc as the heat source to explore the effects of low frequency and arc characteristics on the pore defects and mechanical properties of high-strength Al-Mg aluminum alloy additive manufacturing. The results showed that careful removal of impurities such as oil on the surface of the base metal and wire could reduce the pore nucleation rate from the source. The use of strong and weak pulse period changes to reduce the crystallization rate was beneficial to the escape of hydrogen during the additive process [40].

### 4.3. Crack and Delamination

These defects include high porosity, high residual stress and deformation, cracking, and anisotropic mechanical properties. These are the challenges faced by the wide application of WAAM. Some common defects will be briefly introduced below.

The cracks in WAAM are mainly divided into solidification cracks and liquefaction cracks. The former mainly depends on the material properties, usually because the deposited layer is hindered during the solidification process or caused by high strain in the molten pool. The latter is mainly distributed in the mushy zone or partly melted zone, and part of it during the solidification of the component. The melting zone will be affected by the shrinking force due to the shrinkage of the deposited layer, which may cause liquefaction cracks. Especially in WAAM’s dissimilar metal manufacturing, cracks are more likely to occur due to different material properties.

## 5. Mechanical Properties

The heat input of the arc is higher, and the size of the molten pool and heat-affected zone during the WAAM-forming process is larger. As the surfacing process proceeds, the thermal history of each layer is different. Therefore, studying the crystallographic characteristics and periodicity of formed parts is the basis for controlling WAAM-formed parts.

### 5.1. Hardness Distribution

The Wang Kehong team of Nanjing University of Science and Technology used the HVS1000-Z microhardness tester to measure the hardness value along the center line of the weld along the center line of the weld with the increase in the cladding height and the hardness value at different width positions at the same height when the load was 0.3 kg. As shown in Figure 6, the hardness distribution of the cladding layer had obvious laws: The higher the height, the greater the hardness, and the lower the middle part; the hardness at the center of the weld was high, and the hardness on both sides of the weld was low [61].

### 5.2. Tensile Strength

The Xue Jiaxiang team of South China University of Technology compared the two WAAM processes of high-speed arc welding and high-speed cold pressure welding. The two types of microstructures are shown in Figure 7. Observing the micrographs of the two specimens, it is known that the heat input of high-speed arc welding is high, and the heat accumulation is large, which causes part of the ferrite to melt in the austenite, and the dendrite spacing is large, which also indicates the additive of high-speed arc welding. The tensile strength of the pieces is higher [62].

Xianghui et al. studied the microstructure of additive parts by changing the wire feeding speed, as shown in Figure 8. When the wire feeding speed was increased from 3 to 4 m/min, the arc power also increased. The increase in heat input caused the cooling rate of the back edge of the molten pool to decrease, and the distance between the secondary dendrites increased. Therefore, as the wire feeding speed increased, the dendrite size became larger and the tensile strength decreased [63].

The Feng Yuehai team from Nanjing University of Science and Technology used single-wire and double-wire TIG welding processes to manufacture straight-wall specimens, as shown in Figure 9. Under the same process parameters, the tensile strength improved [64].

The research team of Dalian University of Technology analyzed the influence of laser power on the tensile properties of aluminum alloy additive parts. The result is shown in Figure 10. The laser can increase the tensile strength of the wall, but when the laser power is too high, the tensile strength will decrease [65].

The Wang Kehong team of Nanjing University of Science and Technology studied the mechanical properties of the specimens by increasing the composition of helium in the protective gas. The tensile strength results of the specimens under different protective gas compositions are shown in Figure 11. In the parallel additive direction, the tensile fracture was mainly manifested as ductile fracture; in the vertical additive direction, the tensile fracture was mainly manifested as a mixed-fracture mode of ductile fracture and quasi-cleavage fracture; the tensile strength of the parallel additive direction was stronger than that of the vertical direction [66].

### 5.3. Elongation

The Feng Yuehai team from Nanjing University of Science and Technology used single-wire and double-wire TIG welding processes to manufacture straight-wall specimens, as shown in Figure 12. Under the same process parameters, the elongation after breaking improved [67].

The research team of Dalian University of Technology analyzed the influence of laser power on the tensile properties of aluminum alloy additive parts. The result is shown in Figure 13. The laser can increase the elongation of the wall after breaking, but when the laser power is too high, the elongation will decrease [68].

The Wang Kehong team of Nanjing University of Science and Technology studied the mechanical properties of the specimen by increasing the composition of helium in the protective gas. The results of sample elongation under different protective gas compositions are shown in Figure 14 [46].

It can be seen from the above research that currently focusing on the properties of WAAM specimens, researchers have studied the microstructure, hardness, and tensile fracture morphology, and explored the influence of different processes on the mechanical properties of arc additive components. In view of the complexity of WAAM technology and materials, this direction still needs more in-depth research.

## 6. Current Methods for Quality Improvement in the WAAM Process

### 6.1. Post-Process Heat Treatment

Heat treatment is another strengthening method widely used in WAAM. Heat treatment can effectively reduce residual stress and enhance the mechanical properties of components. However, there are different heat treatment processes for different materials or additive processes, and different heat treatment processes will greatly change the internal structure characteristics and precipitate phases, thereby significantly affecting the structure and performance. Gu [69] explored the effect of T6 heat treatment on the structure and properties of aluminum alloy additive structural parts, and found that after heat treatment, the mechanical properties of aluminum alloy greatly improved, the grain diameter was more uniform, the anisotropy significantly reduced, and the tensile strength and yield strength reached 450 and 305 MPa, respectively, and the investigation found that the main strengthening mechanism was deposition strengthening. Fang et al. [32] compared the effect of heat treatment on the A357 aluminum alloy thin wall and found that after heat treatment, the number of pores in the component was significantly reduced, and the strength and elongation were approximately the same in all directions, which significantly improved the microhardness and intensity. These meet the minimum requirements of castings. However, this technology also has certain drawbacks, which will cause the crystal grains to become coarse, not only increasing the pores but also causing a small number of materials to be destroyed. Therefore, when using the heat treatment process, specific material properties and applications need to be considered.

In 2012, German scholar Brandl [70] used a laser wire-filled additive manufacturing method to make a bulk Ti-6Al-4V titanium alloy, and used two different heat treatment processes to study the changes in structure and mechanical properties. Brandl has also done some exploratory work on the new heat treatment process of additive manufacturing of titanium alloys [71], and proposed a special heat treatment process for the columnar crystal equiaxation phenomenon that appeared in the aforementioned research (Figure 15).

In 2014, domestic scholars Liu et al. [72] explored the heat treatment process for the high strength but poor plasticity of Ti-5Al-5Mo-5V-1Cr-1Fe near-beta titanium alloy under the additive manufacturing state. First, they tried annealing and forging. The standard three-stage heat treatment is commonly used in the state of the material (Figure 16), but the research results show that the former increases the width of the α lath on the β matrix, and the αGB phase continuously distributed along the grain boundary is obviously coarsened, although the plasticity of the material is slightly improved, but both the yield strength and tensile strength decrease.

In 2013, domestic scholars Xu et al. [73] studied the effects of three heat treatment processes on the microstructure and mechanical properties of Inconel625 nickel-based alloys caused by plasma arc rapid prototyping. The experiment carried out direct aging heat treatment, solid solution + aging heat treatment, and homogenization + solid solution + aging heat treatment on the materials in the state of additive manufacturing (Table 3).

### 6.2. Interpass Cold Rolling

Cold rolling, as a plastic processing technology, applies large pressure on the metal surface by rollers to cause large plastic deformation of the material. The rolling deformation can break the coarse columnar crystal structure, significantly increasing the hardness and strength of the deposited layer, and accumulating a large amount of deformation in the material. During the arc deposition of the next layer, when the heat of the arc deposition reaches the recrystallization temperature of the material, it will induce the recrystallization process of the material and refine the crystal grains, thereby improving the mechanical properties of the material and improving the anisotropy. As rolling deformation can produce a greater deformation force than ultrasonic impact, Cranfield University researchers introduced cold rolling deformation into the arc additive process to achieve arc deposition and cold rolling. The rolling composite manufacturing principle is as follows, shown in Figure 17.

The cold rolling process adopted by Cranfield University is to apply cold rolling deformation to the deposited layer after the arc deposits a layer and when the temperature of the deposited layer drops to room temperature. The rolling force is controlled by a hydraulic pressing device to achieve different pressures. Colegrove et al. and Donoghue et al. studied the influence of the cold rolling process and different rolling forces on β grain refinement and the texture of TC4 titanium alloy. As the rolling pressure increases, the grain refinement effect increases and the texture heterogeneity decreases [74,75]. Martina further studied the effect of different rolling pressures on the microstructure of TC4 titanium alloy during the cold rolling process, and found that when the cold rolling pressure reached 75 kN, almost all the internal columnar crystals transformed into equiaxed microstructures [76]. Mcandrew et al. studied the influence of rolls with different cross-sectional characteristics on cold rolling deformation and found that rolls with grooves or protrusions can produce larger and deeper deformations than flat rolls [32].

### 6.3. Interpass Cooling

In 2012, German scholar Brandl [55] used the laser wire-filled additive manufacturing method to make a bulk Ti-6Al-4V titanium alloy, and used two different heat treatment processes to study the changes in the structure and mechanical properties. It was found that the structure of the Ti-6Al-4V titanium alloy in the additive manufacturing state was directional growth of coarse β grains, accompanied by banding characteristics in the macroscopic view. After aging heat treatment at 600 °C for 4 h and cooling with the furnace, the structure characteristics hardly changed, but the hardness significantly improved, and the distribution of Al and V elements in the alloy tended to be uniform. After the solution treatment with 1200 °C for 2 h and furnace cooling, the columnar β grains were equiaxed, resulting in an effect similar to recrystallization, the band characteristics disappeared, but the hardness decreased.

In 2010, domestic scholar Wang et al. [77] studied the effect of heat treatment on the microstructure and mechanical properties of deposited martensitic stainless steel 1Cr12Ni2WMoVNb. The stainless steel was formed from prefabricated powder by coaxial feeding by laser melting and deposition. Each was at 1050 °C, 1100 °C, 1150 °C, and 1200 °C; temperature solution treatment was performed for 30 min and then quenched in oil; and then the 1150 °C solution treatment sample was quenched and tempered (heated to 580 °C, tempered and cooled in air).

In 2016, Australian scholar Ma et al. [78] used arc additive manufacturing technology to prepare γ-TiAl intermetallic compounds and studied the effect of heat treatment on its structure and mechanical properties. Two heat treatment schemes were implemented, which were kept at 1060 °C and 1200 °C for 24 h and then cooled with the furnace. The experiment found that the microstructure change of the structure obtained by the former was a lamellar structure with different sizes, while the latter obtained equiaxed gamma grains. Changes in mechanical properties, on the one hand, with heat treatment at 1060 °C increased the strength index of the material, but the plasticity was lost; heat treatment at 1200 °C significantly improved the plasticity, so the anisotropy of mechanical properties almost disappeared, but the strength index decreased slightly.

### 6.4. Peening and Ultrasonic Impact Treatment

Ultrasonic impact is a plastic forming and strengthening process. In ultrasonic impact, the ultrasonic transducer and horn transmit the high-frequency vibration of the ultrasonic generator to the surface of the workpiece through the impact tool head, forming a plastic deformation layer on the surface of the workpiece. The plastic deformation caused by high-frequency impact can effectively break the coarse columnar crystals formed during the arc additive process to form fine equiaxed crystal structures, thereby improving the mechanical properties and uniformity of materials. In addition, the impact will form a residual compressive stress layer on the surface of the material, which can significantly reduce the harmful residual tensile stress introduced by the arc additive process, and at the same time reduce the warpage and deformation of the parts after the arc additive manufacturing and the internal microscopic void defects.

Zhi et al. studied the influence of ultrasonic shock on the mechanical properties of arc-added titanium alloy parts. As shown in Figure 18, ultrasonic shock can reduce the anisotropy of mechanical properties of parts, and the percentage of anisotropy in tensile strength of parts was 12.5%, reduced by 1.5% [79].

Donoghue et al. studied the effect of ultrasonic shock on the microstructure of arc-added TC4 titanium alloy, and found that ultrasonic shock can break coarse columnar crystals and form fine equiaxed crystal structures. However, even in each layer of arc increase, after applying ultrasonic shock to the material, there were still some unbroken columnar crystal structures. From the above analysis, different impact energies and impact times have a significant impact on the microstructure of the metal.

## 7. Conclusions and Prospects

Although WAAM technology is widely used, it has problems such as a difficult-to-control precision of the formed parts and poor organization performance. In recent years, domestic and foreign scholars’ research on WAAM technology has mainly focused on the following directions:(1)WAAM complex component manufacturing based on offline programming;(2)Monitoring and forming control of WAAM process;(3)The influence of different process methods and process parameters on the macroscopic and microstructure properties of additive parts and the optimization of process parameters.

Compared with traditional additive processing technology, WAAM technology has more prominent technical advantages and application prospects, and will be applied in more fields. WAAM trajectory planning and optimization, arc additive composition monitoring, WAAM process optimization and forming control, and other issues will be the focus of attention in the future.

This paper analyzes and summarizes the current research status of arc additive manufacturing technology, and has a comprehensive and systematic understanding of the current research and development status of this technology field. This summary can provide a guiding direction for the subsequent research on arc additive manufacturing technology. Moreover, at this stage, my country’s manufacturing industry is developing rapidly, and the requirements for manufacturing efficiency and product quality are getting higher and higher. In this context of development, the exploration and use of emerging manufacturing technologies are particularly important. It is believed that arc additive manufacturing technology will be fully promoted and applied in the future manufacturing industry by virtue of its high efficiency and automation advantages.

## Figures and Tables

**Figure 1 materials-14-01415-f001:**
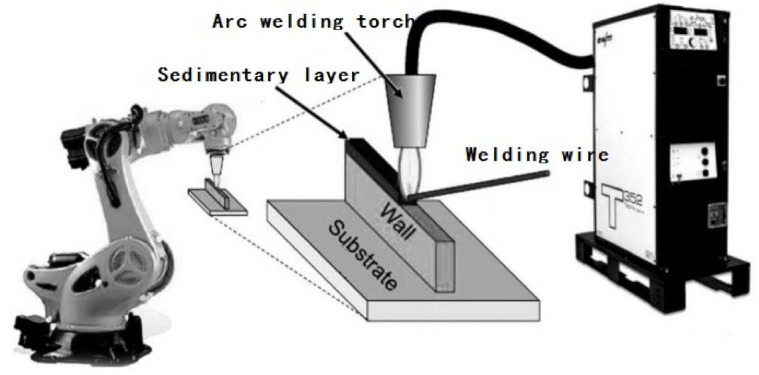
WAAM technology process.

**Figure 2 materials-14-01415-f002:**
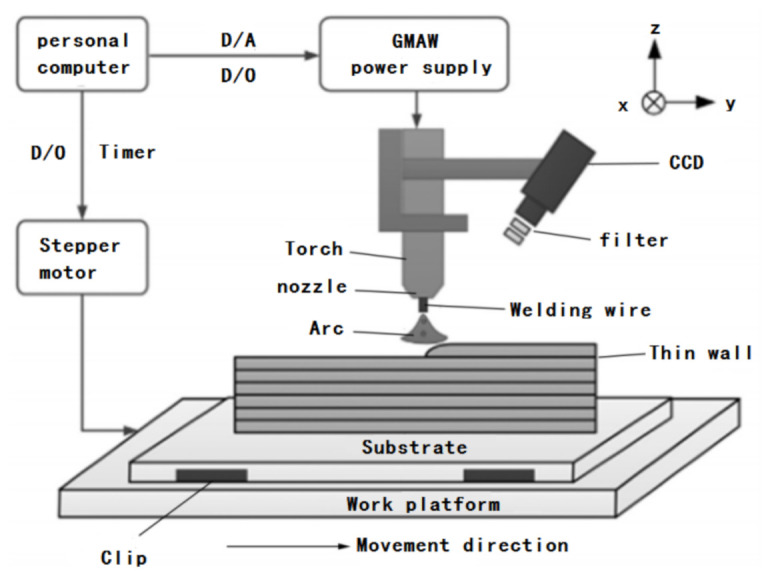
Gas metal arc welding (GMAW) additive manufacturing schematic diagram [9].

**Figure 3 materials-14-01415-f003:**
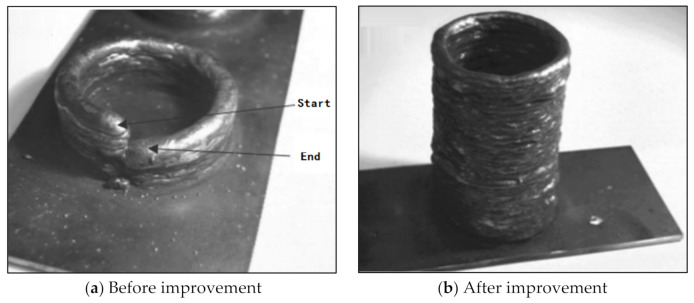
Tubular parts.

**Figure 4 materials-14-01415-f004:**
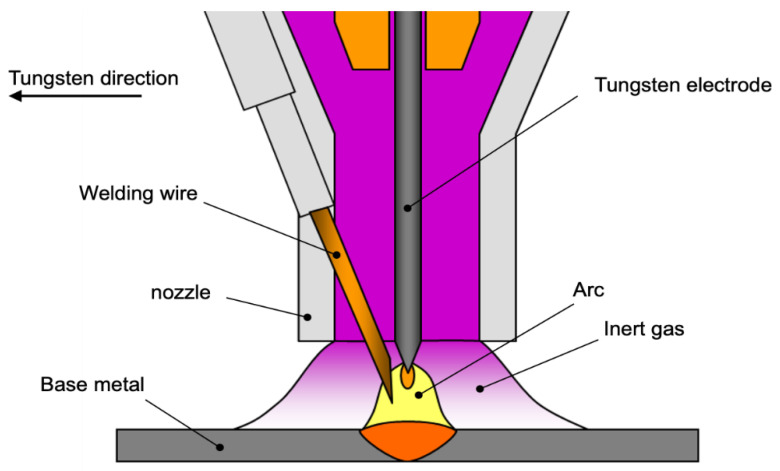
Principle of gas tungsten arc welding (GTAW) surfacing.

**Figure 5 materials-14-01415-f005:**
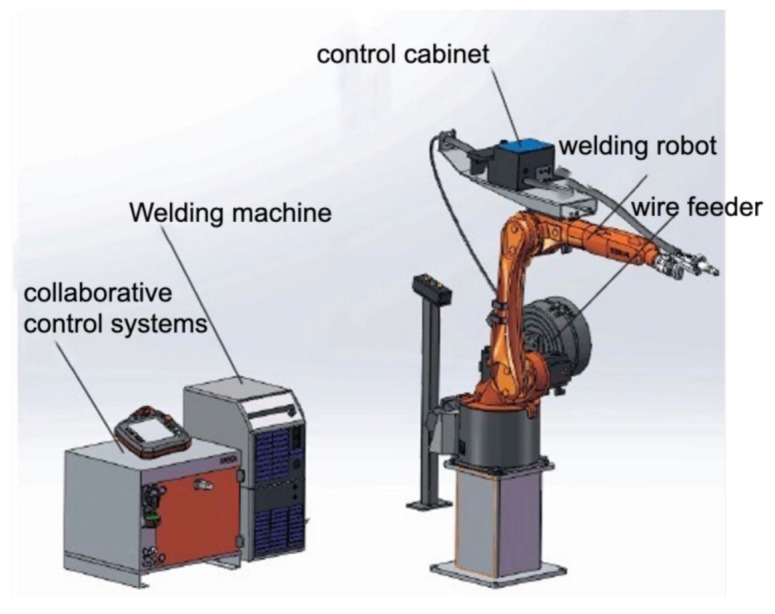
Robot WAAM system.

**Figure 6 materials-14-01415-f006:**
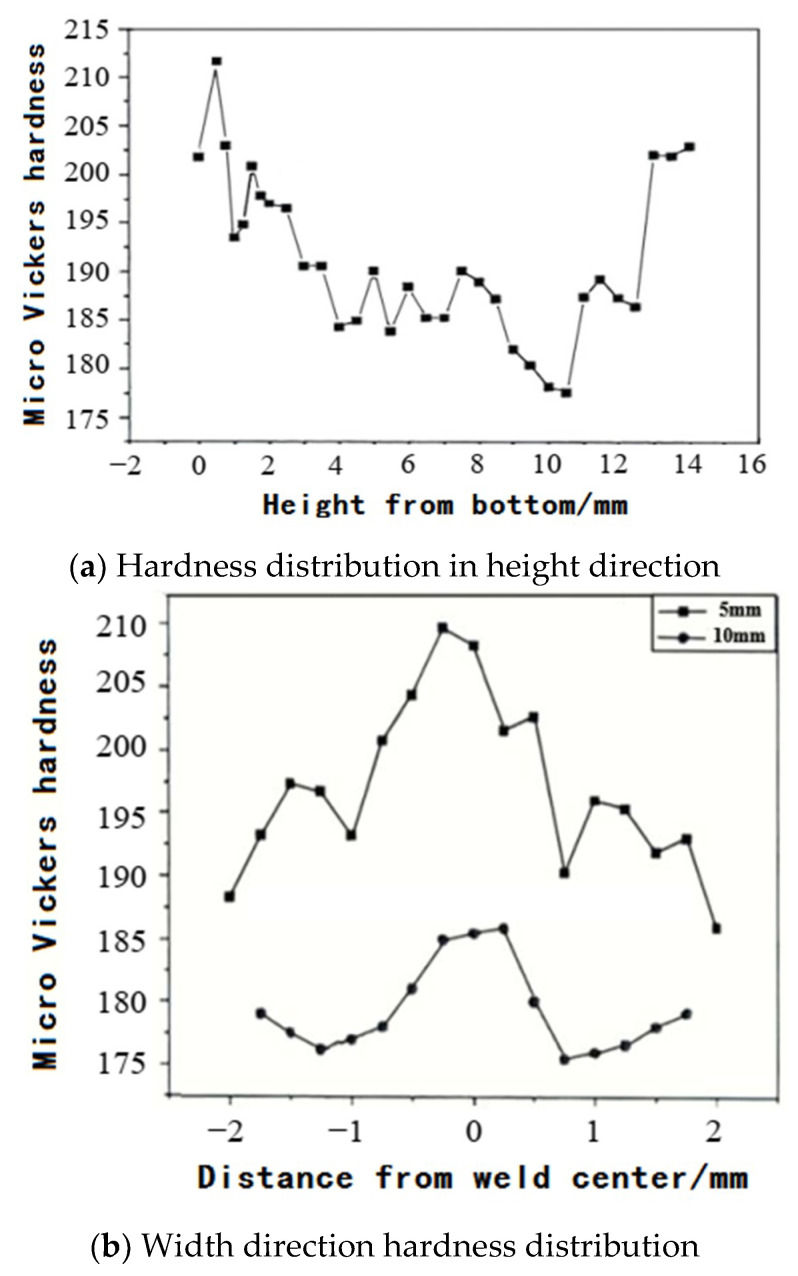
Hardness distribution diagram.

**Figure 7 materials-14-01415-f007:**
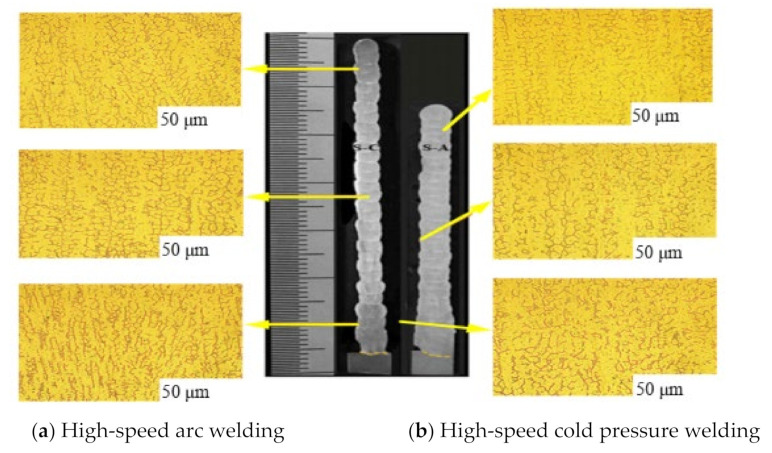
Microscopic view of two welding methods.

**Figure 8 materials-14-01415-f008:**
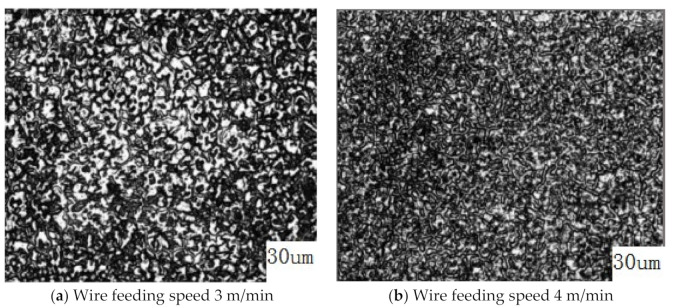
Metallographic organization chart at different wire feeding speeds.

**Figure 9 materials-14-01415-f009:**
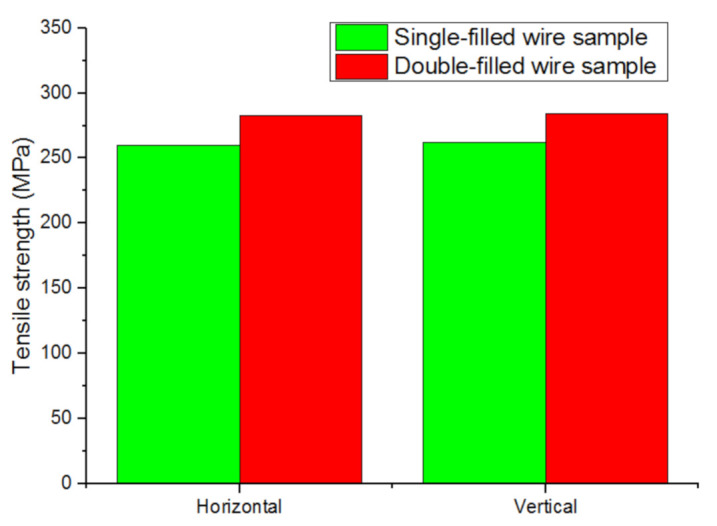
Tensile strength of single-filled and double-filled samples.

**Figure 10 materials-14-01415-f010:**
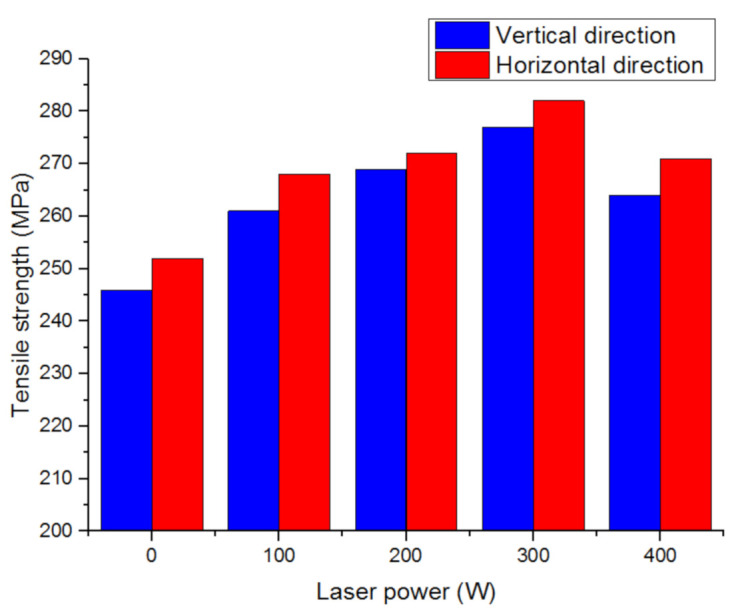
The influence of different laser powers on tensile strength.

**Figure 11 materials-14-01415-f011:**
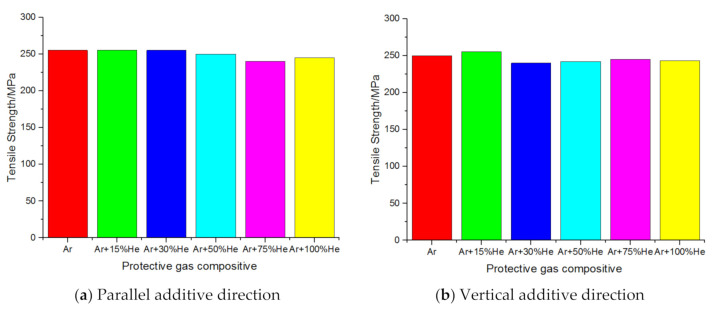
Tensile strength of samples under different protective gas compositions.

**Figure 12 materials-14-01415-f012:**
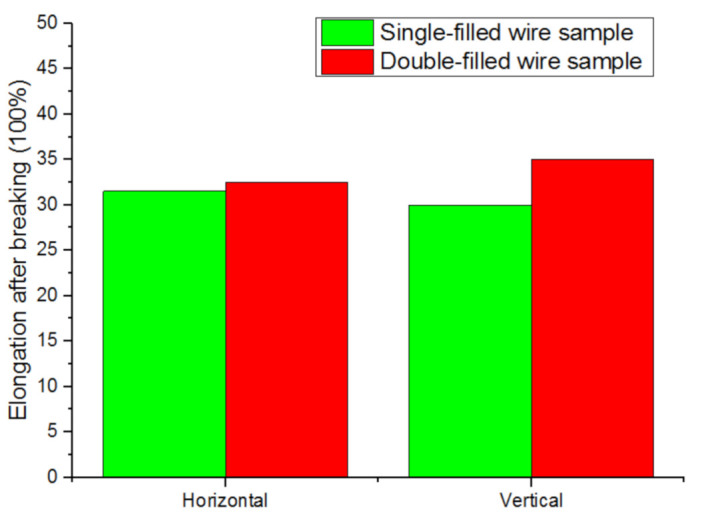
Elongation of single-filled wire and double-filled wire samples.

**Figure 13 materials-14-01415-f013:**
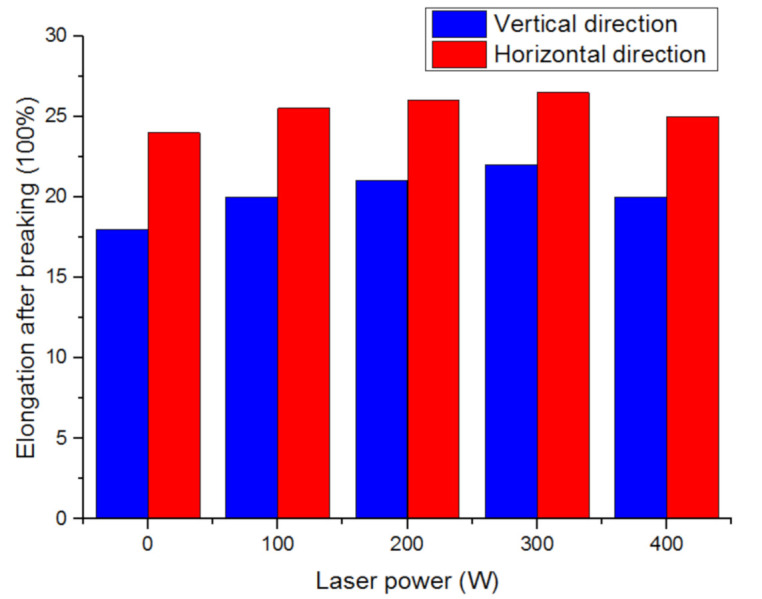
Effect of different laser powers on elongation after breaking.

**Figure 14 materials-14-01415-f014:**
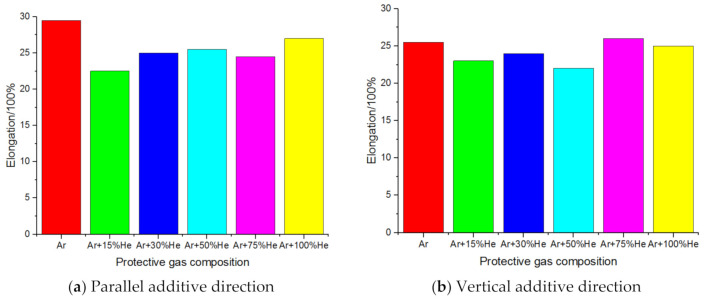
Sample elongation under different protective gas compositions.

**Figure 15 materials-14-01415-f015:**
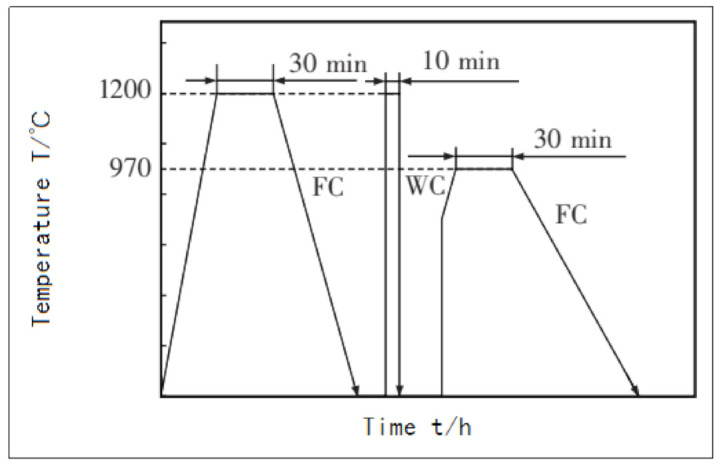
Heat treatment process.

**Figure 16 materials-14-01415-f016:**
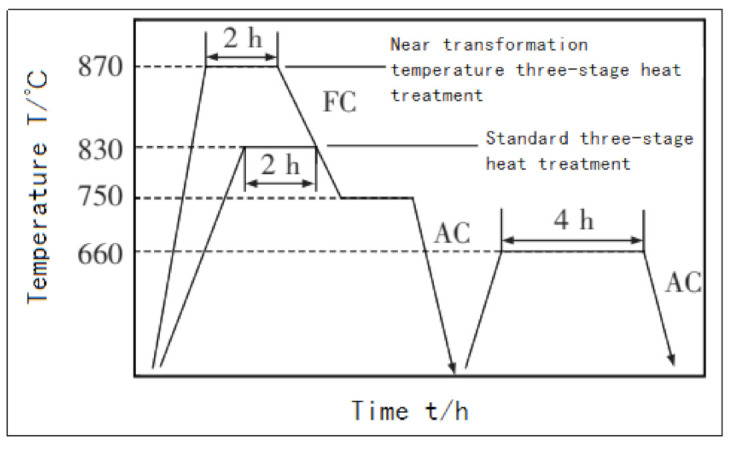
Heat treatment process curve.

**Figure 17 materials-14-01415-f017:**
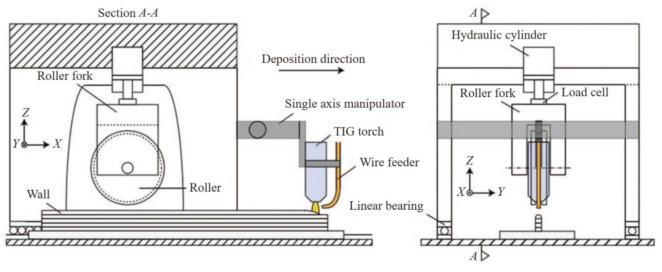
Schematic diagram of cold rolling deformation and arc additive manufacturing technology.

**Figure 18 materials-14-01415-f018:**
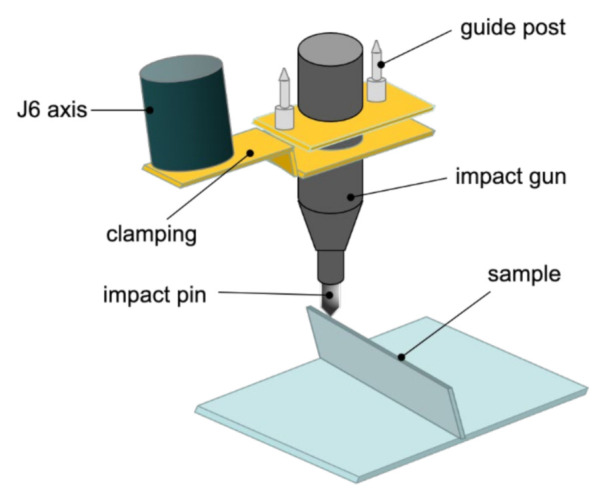
Schematic diagram of WAAM and hybrid ultrasonic peening.

**Table 1 materials-14-01415-t001:** Research overview of wire arc additive manufacturing (WAAM) technology in China and abroad.

Research Institute	Molding Material	Molding	Research Content	References
Cranfield University	Ti–6Al–4V	TIG	The effect of interlayer rolling on WAAM residual stress	[1]
University of Firenze	——	GMAW	WAAM modeling analysis based on a new heat source model	[2]
University of Manitoba	ATI 718Plus	TIG	Microstructure analysis of ATI 718 Plus alloy WAAM	[3]
Southern Methodist University	5356 welding wire	GTAW	Forming 5356 aluminum alloy parts by variable-polarity tungsten argon arc welding	[4]
Huazhong University of Science and Technology	ERTi-5	PAW	Research on Microstructure and Properties of Ultrasonic Impact WAAM Titanium Alloy Parts	[5]
Beijing University of Aeronautics and Astronautics	ER-2319	TIG	The influence of laser shock strengthening on WAAM microstructure and residual stress	[6]
Harbin Institute of Technology	Inconel625	GTAW	Research on WAAM Process of Inconel 625 Alloy	[7]
Tianjin University	5356 welding wire	MIG	The influence of welding parameters and path on the size of deposited layer	[8]

**Table 2 materials-14-01415-t002:** Types of main robot welding platform.

Welding Platform Type			
KUKA six-degrees-of-freedom robot arc welding platform	ABB six-axis robotic arc welding platform	FANUC six-axis robotic arc welding platform	YASAKAWA MOTOMAN robot arc welding platform

**Table 3 materials-14-01415-t003:** Mechanical properties of materials before and after heat treatment [73].

Status	Tensile Strength/MPa	Elongation After Breaking A (%)	Yield Strength/MPa
Sedimentary state	771	50	480
Direct aging	833	38	495
Solid solution + aging	851	44	535
Homogenization + solid solution + aging	732	40	449
forging	855	50	490

## Data Availability

As this is a review paper, no new data was generated for this paper.

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
