# Peer review of "Research Progress of Arc Additive Manufacture Technology"

_materials, 2021, doi:10.3390/ma14061415_

Round 1

Reviewer 1 Report

This article expounds the domestic and foreign research and application status of Wire Arc Additive Manufacture (WAAM) technology from several major perspectives such as the control of defects such as forming system, forming material, residual stress and pores, as well as mechanical properties and process quality improvement.

This research is relevant and interesting because presents the advantages of WAAM technology, comparing with other additive manufacturing technologies. It is showed different WAAM systems that are also interesting for readers.

The paper is well written. Authors present the development of processing method and application of composite energy fields. Different metals which can be used in WAAM process are showed.

The conclusions are consistent with the evidence and arguments presented.

Specific Comments:

  1. Please, write the Celsius degrees with space (lines 105, 304, 305, 382, 508, 752, 754, 765, 769, 770).
  2. Please, modify "4 hours" with "4h".
  3. Correct the References using the Guide of the Journal.

Author Response

Response 1: Please write Celsius degrees with space(lines 105, 3 04, 305, 382, 508, 752, 754, 765, 770).

I have revised  all the Celsius degrees with space .

Response 2: Please, modify "4 hours" with "4h".

I have revised  "4 hours" with "4h".

Response 3: Correct the References using the Guide of the Journal.

I have revised  the format of  references.

Reviewer 2 Report

In my opinion, work generally requires a few adjustments:

  1. Fig 2. in the caption I would suggest including the literature from which it was taken.
  2. Why are the footnotes in the text written in superscript?References to the literature appear in the text in [..], while in the literature list1.…
  3. Fig 6. No dimension line in the drawing referring to 0.5?
  4. 7. The scale is marked only for (b), while in the remaining figures there is no scale, please complete. The caption lacks a description of what the individual drawings show,
  5. Missing figure in the caption, as shown in the individual figures (a÷f),
  6. Various dimensions and non-uniform descriptions of the axes in Figure 9,
  7. 11 no dimension - scale (dimension designation)
  8. Fig 12. Poor drawing quality, no measurement errors marked,
  9. Figures 12 and 13 as well as 14 and 15 it would be good to present uniformly, as they present analogous results
  10. Figs 16 and 17 correct the descriptions of the individual columns on the horizontal axis. They are poorly visible and shifted in relation to the described columns.
  11. 23. No scale,
  12. Please correct the abstract, which is to be a short description of the work's content and not an introduction.
  13. In the redrafted work, please indicate what is the contribution of the authors of the work to the topic of the work,
  14. In the section "conclusions and perspectives, conclusions are missing in my opinion. This part of the work is too general.
  15. Literature review should be adapted to the requirements of the Materials Journal.

Author Response

Thank you for your help. I have revised the comments.

Response 1: Fig 2. in the caption I would suggest including the literature from which it was taken.

I've added literature quoted [9].

Response 2:Why are the footnotes in the text written in superscript?References to the literature appear in the text in [..], while in the literature list1.…

I have revised  the format of  references.

Response 3: Fig 6. No dimension line in the drawing referring to 0.5?

I have deleted the pictiure because of copyrights.

Response 4: The scale is marked only for (b), while in the remaining figures there is no scale, please complete. The caption lacks a description of what the individual drawings show.

I have deleted the pictiure because of copyrights.

Response 5: Missing figure in the caption, as shown in the individual figures (a÷f),

I have drawn the referring.

Response 6: Various dimensions and non-uniform descriptions of the axes in Figure 9,

I have drawn the referring in Figure 8.

Response 7: 11 no dimension - scale (dimension designation)

I have drawn the referring.

Response 8: Fig 12. Poor drawing quality, no measurement errors marked.

I have redrawn the figure.

Response 9: Figures 12 and 13 as well as 14 and 15 it would be good to present uniformly, as they present analogous results。

I have redrawn the figures.

Response 10: Figs 16 and 17 correct the descriptions of the individual columns on the horizontal axis. They are poorly visible and shifted in relation to the described columns.

I have redrawn the figures.

Response 11: 23. No scale,

I have deleted the pictiure because of copyrights.

Response 12: Please correct the abstract, which is to be a short description of the work's content and not an introduction.

I've added the description of abstract.

Response 13: In the redrafted work, please indicate what is the contribution of the authors of the work to the topic of the work,

I've added the description of abstract.

Response 14: In the section "conclusions and perspectives, conclusions are missing in my opinion. This part of the work is too general.

I've added the description of conclusions.

Response 15: Literature review should be adapted to the requirements of the Materials Journal.

I have revised  the format of  references.

Reviewer 3 Report

In this manuscript, the authors expounded the domestic and foreign research and application status of WAAM technology from several major perspectives such as the control of defects such as forming system, forming material, residual stress and pores, as well as mechanical properties and process quality improvement.

Comments:

  1. The article has a large number of grammatical errors, typos and requires serious corrections.
  2. The article focuses on the consideration of domestic and foreign research as a material for the review of WAAM technology. Proceeding from the fact that the authors of the work are representatives of three countries, it is not clear what is meant by domestic and foreign research and whether it is necessary to focus on this.
  3. A lot of article close connected with the main theme of the your article have not realized. I think that authors need check more article. For example, they can read: https://doi.org/10.3390/ma13112491; https://doi.org/10.1134/S102745101901004X; https://doi.org/10.1016/j.apsusc.2019.05.068.
  4. Affiliation 4 is not associated with any of the authors.
  5. Section 4 requires revision in terms of structure, as its last subsection 4.3 contains a sentence: “Although there are many advantages of WAAM, there are still some defects in WAAM manufacturing that need attention and resolution.”, which would be more logical put at the beginning of the section.
  6. Typical typos, of which there are a lot in this work:

Line 56 "of GTAW, GTAW or PAW."

Line 78 "roughness, To study"

Line 120 "but the equipment It is"

Line 122-123 "welding, penetration plasma arc welding, and penetration plasma arc welding."

Line 128 "layer were the coarse"

Line 138 "structure is simple For small"

Line 139 "The pulse plasma arc welding plasma arc column has"

Line 140 "Therefore, researchers are paying more and more attention." The subject is missed.

Line 171 "finally solidifies Molding to"

Line 189 "Rolls Royce (Roll-Royce)."

Line 197 "the material During the stacking"

Line 264 “There are differences [21,22] .F. Wang et al. " ???

And so on ...

7. Poor image quality. There is no scale in the figures.

8. It is not clear what the authors wanted to say with this sentence: “In short, how to suppress the formation of undesirable structures inside the molded parts in the arc additive preparation of titanium alloys, effectively control the evolution of grains and microstructures during the layer-by layer accumulation process, reduce anisotropy, and improve the comprehensive mechanical properties of parts has become the follow-up research in this area. important topic [34].

9. And thus, line 358: “of the material. And plasticity. As the welding… "

10. It is not clear why section 3.3 contains information about nickel alloy powders, if the article deals with wire-arc additive technology: “Nickel-based alloy powders for material manufacturing require high purity (low content of impurities such as nitrogen, oxygen, and sulfur), good sphericity (to maintain .m, good fluidity), and narrow particle size distribution (the particle size is generally 5-150 and the middle can be Subdivision). At the same time, nothing is said about the use of nickel wires for WAAM.

11. There is no reference to figure 7 in the text, its use is questionable, as well as the use of figure 8.

12. Section 4.2 provides information about CMT, but nowhere is a transcript of this abbreviation or description of this technology given. "Compared with the other three modes, the traditional CMT mode produces the most pores, with some pores exceeding 100.m in diameter."

Conclusion: the article requires significant corrections.

Author Response

Thank you for your time to help me,I have revised my paper for your comments

Line 56 "of GTAW, GTAW or PAW."

I have revised the grammatical errors in line 67.

Line 78 "roughness, To study"

I have revised the grammatical errors in line 90.

Line 120 "but the equipment It is"

I have revised the grammatical errors in line 134.

Line 122-123 "welding, penetration plasma arc welding, and penetration plasma arc welding."

I have revised the grammatical errors in line 136.

Line 128 "layer were the coarse"

I have revised the grammatical errors in line 142.

Line 138 "structure is simple For small"

I have revised the grammatical errors in line 151.

Line 139 "The pulse plasma arc welding plasma arc column has"

I have revised the grammatical errors in line 153.

Line 140 "Therefore, researchers are paying more and more attention." The subject is missed.

I have revised the grammatical errors in line 163.

Line 171 "finally solidifies Molding to"

I have revised the grammatical errors in line 186.

Line 189 "Rolls Royce (Roll-Royce)."

I have revised the grammatical errors in line 205.

Line 197 "the material During the stacking"

I have revised the grammatical errors in line 213.

Line 264 “There are differences [21,22] .F. Wang et al. " ???

I have revised the grammatical errors in line 284.

  1. Poor image quality. There is no scale in the figures.

I have redrawn the figures.

  1. It is not clear what the authors wanted to say with this sentence: “In short, how to suppress the formation of undesirable structures inside the molded parts in the arc additive preparation of titanium alloys, effectively control the evolution of grains and microstructures during the layer-by layer accumulation process, reduce anisotropy, and improve the comprehensive mechanical properties of parts has become the follow-up research in this area. important topic [34].

I have revised the grammatical errors in line 335.

  1. And thus, line 358: “of the material. And plasticity. As the welding… "

I have revised the grammatical errors in line 302.

  1. It is not clear why section 3.3 contains information about nickel alloy powders, if the article deals with wire-arc additive technology: “Nickel-based alloy powders for material manufacturing require high purity (low content of impurities such as nitrogen, oxygen, and sulfur), good sphericity (to maintain .m, good fluidity), and narrow particle size distribution (the particle size is generally 5-150 and the middle can be Subdivision). At the same time, nothing is said about the use of nickel wires for WAAM.

I have deleted the part ‘3.3 Nickel-based superalloys’.

  1. There is no reference to figure 7 in the text, its use is questionable, as well as the use of figure 8.

I have deleted the figure.

  1. Section 4.2 provides information about CMT, but nowhere is a transcript of this abbreviation or description of this technology given. "Compared with the other three modes, the traditional CMT mode produces the most pores, with some pores exceeding 100.m in diameter."

I have added the description of CMT in line 566.

Round 2

Reviewer 3 Report

The authors took into account all comments. No new comments.